# Genome-Wide Identification of *Eucalyptus* Heat Shock Transcription Factor Family and Their Transcriptional Analysis under Salt and Temperature Stresses

**DOI:** 10.3390/ijms23148044

**Published:** 2022-07-21

**Authors:** Tan Yuan, Jianxiang Liang, Jiahao Dai, Xue-Rong Zhou, Wenhai Liao, Mingliang Guo, Mohammad Aslam, Shubin Li, Guangqiu Cao, Shijiang Cao

**Affiliations:** 1College of Forestry, Fujian Agriculture and Forestry University, Fuzhou 350002, China; ytan0826@163.com (T.Y.); jxliang_china@126.com (J.L.); daijiahao123@126.com (J.D.); lwh1623850793@163.com (W.L.); fjlishubin@126.com (S.L.); 2Commonwealth Scientific Industrial Research Organization (CSIRO) Agriculture Food, Canberra, ACT 2601, Australia; xue-rong.zhou@csiro.au; 3Center for Genomics and Biotechnology, Fujian Agriculture and Forestry University, Fuzhou 350002, China; gml604@163.com (M.G.); aslampmb1@gmail.com (M.A.); 4College of Plant Protection, Fujian Agriculture and Forestry University, Fuzhou 350002, China

**Keywords:** heat shock transcription factor, *Eucalyptus grandis*, biological analysis, gene expression, abiotic stress

## Abstract

Heat shock transcription factors (HSFs) activate heat shock protein gene expression by binding their promoters in response to heat stress and are considered to be pivotal transcription factors in plants. *Eucalyptus* is a superior source of fuel and commercial wood. During its growth, high temperature or other abiotic stresses could impact its defense capability and growth. *Hsf* genes have been cloned and sequenced in many plants, but rarely in *Eucalyptus*. In this study, we used bioinformatics methods to analyze and identify *Eucalyptus* *Hsf* genes, their chromosomal localization and structure. The phylogenetic relationship and conserved domains of their encoded proteins were further analyzed. A total of 36 *Hsf* genes were identified and authenticated from *Eucalyptus*, which were scattered across 11 chromosomes. They could be classified into three classes (A, B and C). Additionally, a large number of stress-related cis-regulatory elements were identified in the upstream promoter sequence of *HSF*, and cis-acting element analysis indicated that the expression of *EgHsf* may be regulated by plant growth and development, metabolism, hormones and stress responses. The expression profiles of five representative *Hsf* genes, *EgHsf4*, *EgHsf9*, *EgHsf13*, *EgHsf24* and *EgHsf32*, under salt and temperature stresses were examined by qRT-PCR. The results show that the expression pattern of class B genes (*EgHsf4*, *EgHsf24* and *EgHsf32*) was more tolerant to abiotic stresses than that of class A genes (*EgHsf9* and *EgHsf13*). However, the expressions of all tested *Hsf* genes in six tissues were at different levels. Finally, we investigated the network of interplay between genes, and the results suggest that there may be synergistic effects between different *Hsf* genes in response to abiotic stresses. We conclude that the *Hsf* gene family played an important role in the growth and developmental processes of *Eucalyptus* and could be vital for maintaining cell homeostasis against external stresses. This study provides basic information on the members of the *Hsf* gene family in *Eucalyptus* and lays the foundation for the functional identification of related genes and the further investigation of their biological functions in plant stress regulation.

## 1. Introduction

Forest trees benefit people by removing carbon from the atmosphere, cooling our surroundings and filtering our water and air, as well as providing us with shelter, livelihoods, water, food and fuel security [1,2]. As an important forest tree, *Eucalyptus* has great research value. *Eucalyptus* is the general name of *Eucalyptus grandis*, in the Myrtaceae family. It is known worldwide as one of the most famous species due to its fast growth rate, high economic value and great ecological and social benefits [3]. As an excellent fuel wood and commercial tree species, *Eucalyptus* is widely used for plantation forestry in tropical and subtropical regions [4]. The growth and development of *Eucalyptus* are often affected by abiotic stresses such as temperature, water and salt. High temperature and drought can cause chlorophyll degradation, damage chloroplast membranes and reduce photosynthetic efficiency [5]. These abiotic stresses may occur repeatedly and gradually over time in natural environments. Therefore, they can lead to some response mechanisms in plants through genetic regulation, such as “temperature memory” in response to temperature [6]. Further exploration of these mechanisms lies in the study of key regulatory genes. Transcription factors (TFs) are important regulatory proteins in plants and animals. TFs can precisely regulate the spatiotemporal-specific expression of downstream genes, which are also key factors in regulating growth and stress responses [7]. Researchers have identified many transcription factors, some of which are involved in the heat shock response [8]. The heat shock transcription factor is a key transcription factor responding to heat stress. Not only does it play a vital role in receiving and transmitting heat signals, regulating downstream gene expression, stimulating resistance responses, transferring heat conduction stress and producing heat resistance, but it also participates in regulating protein distribution, degradation and folding [9,10,11].

Plants need to adapt to changing environmental conditions, especially abiotic stresses [12]. Heat shock protein (HSP) is a specific protein in plants with a large replicative capacity that contributes to the recovery of cell structure and functional reconstruction [13]. The significance of HSF lies in its powerful ability to maintain protein stability under stressful circumstances [14,15]. Responding to heat and other stresses, HSF plays a vital role in protecting normal growth and maintaining cell homeostasis [16,17,18,19]. In addition to responding to high temperature, HSF can also help plants to cope with other adverse conditions, such as drought and cold [20]. The heat shock motif of the *Hsp* gene promoter can be specifically recognized and bound by HSF, thus resulting in the mediation and activation of *Hsp* gene expression. Therefore, HSF is the central regulator of the heat stress response. *Hsf* genes are clustered into five conserved classes [21]. Furthermore, typical HSFs consist of several structurally and functionally conserved domains, including the DNA-binding domain (DBD), the N-terminal adjacent bipartite oligomerization domains (HR-A/B), the nuclear localization signals (NLS), the nuclear export signal (NES) and, in some cases, the C-terminal transcriptional activation domain (CTAD) [19,22]. In addition, HSFs in plants can be divided into at least three classes by a flexible linker of variable lengths, with approximately 15~80 amino acid residues and an HR-A/B region, namely class A (containing nine subclasses, A1 to A9), class B (containing four subclasses, B1 to B4) and class C (containing two subclasses, C1 and C2) [22,23]. In recent years, there has been an explosion of genome-wide plant studies that can fully improve our understanding of the genetic basis of plant diversity and provide a large spectrum of genes and variants associated with important traits and environmental adaptations [24,25,26,27,28]. The identification of *Hsf* family members is the basis of studying the principle of the plant thermal response at the gene level. Through genome-sequencing technology, *Hsf* gene families have been identified in various plants, such as wheat, grape, pepper, moso bamboo and cucumber [29,30,31,32,33]. Recently, whole-genome sequencing has been completed for many plants, such as *Arabidopsis*, rice, tomato, corn and poplar [34,35,36,37,38]. Despite the fact that *Hsf* families have been comprehensively studied in other species [19,29,39,40], in-depth research on the *Hsf* family in *Eucalyptus* has not yet been reported. In this study, the *Hsf* gene family of *Eucalyptus* was identified at the genome-wide level, and systematic bioinformatics analysis was performed to provide a basis for the further identification and cloning of related genes in *Eucalyptus*.

## 2. Results

### 2.1. Identification of EgHsfs in Eucalyptus

Homologous comparison was conducted in the HSF transcription factor family, and a total of 36 proteins with HSF characteristics were identified by SMART validation after removing the repetitive sequences. The 36 corresponding *EgHsf* genes were classified: 26 *EgHsf* members belonged to class A (*EgHsfA*), 9 *EgHsf* members belonged to class B (*EgHsfB*), and 1 *EgHsf* member belonged to class C (*EgHsfC*).

*EgHsf30* and *EgHsf32* were the two longest gene members with 6363 nt and 5275 nt, respectively (Table 1 and Appendix A). The former encoded a protein with a pI of 5.93 and a theoretical MW of 54,320 Da. The latter encoded a protein with a pI of 6.68 and an MW of 46,303 Da. The length of other *EgHsf* members ranged from 1349 to 4442 nt. From the perspective of pI value, EgHSF26 and EgHSF34 were on the high side, being 9.41 and 9.33, respectively. There were four sequences, EgHSF29, EgHSF3, EgHSF28 and EgHSF33, with pI values between 7 and 8. The remaining proteins had pI values between 5 and 6. This result suggests that most HSF proteins were negatively charged, and only a few were positively charged.

### 2.2. Phylogenetic Analysis and Classification of EgHSF Protein Members

In order to study the evolutionary characteristics of genes and the evolutionary relationships among HSF proteins, we constructed a phylogenetic tree (Figure 1) with 36 *Eucalyptus* HSF proteins, along with 21 and 25 HSF proteins from *Arabidopsis* and rice, respectively. According to the known *Arabidopsis* and rice HSF families, EgHSF could be divided into three classes, A, B and C. Among them, class A was the largest, with 26 members. These classes could be further divided into nine subclasses, EgHSFA1–9, in which the EgHSFA9 subclass contained 16 members [19], including EgHSF7, EgHSF10, EgHSF11, EgHSF12, EgHSF13, EgHSF14, EgHSF15, EgHSF16, EgHSF17, EgHSF18, EgHSF19, EgHSF20, EgHSF21, EgHSF22, EgHSF23 and EgHSF36. These members shared a higher homology with AtHSFA9 from *Arabidopsis* based on the phylogenetic tree. Class B contained nine members, including EgHSF3, EgHSF24, EgHSF26, EgHSF28, EgHSF29, EgHSF32, EgHSF33 and EgHSF34, which could be further divided into EgHSFB1–4 subclasses. Class C contained only one member (EgHSF8).

### 2.3. Chromosomal Distribution and Synteny Analysis of the EgHsf Genes

We further analyzed the *EgHsf* gene location on the *Eucalyptus* chromosomes (Appendix A). It was clear that the distribution of *Hsf* genes across 11 chromosomes was uneven (Figure 2). As many as eighteen genes were located on chromosome 3, accounting for up to half of the total number of genes, while five genes were located on chromosome 1. The numbers of genes distributed on other chromosomes were mostly between one and two, except for three genes on chromosome 5 and no gene on chromosomes 2 and 7. Only one gene, *EgHsf*36, was not located on any chromosome, probably due to the genome annotation. We found that most of the genes of the *EgHsfA* subfamily were located on chromosome 3, while *EgHsf**B* and *EgHsf**C* subfamily genes were discretely distributed on other chromosomes.

A tandem repeat event is generally defined when two or more genes are contained within a 200 kb region on a chromosome [41]. Interestingly, we found that there were 14 *EgHsf* genes on chromosome 3 of the *Eucalyptus* linkage group (*EgHsf10* to *EgHsf23*), forming a relatively large tandem duplication region; it thus seemed that chromosome 3 was a hot spot for studying the distribution of *EgHsf* genes. In addition, there were five fragment repeats located at both ends or near both ends of chromosomes 1, 3, 5, 8 and 11, respectively.

In order to further study the gene duplication, we constructed seven *EgHsf* comparative collinear maps (Figure 3). Among them, the strength of the correlation with *EgHsf* genes in descending order was as follows: poplar (42), tomato (29), grape (27), corn (26), *Arabidopsis* (21), rice (18) and barley (10). Comparative collinear mapping between monocotyledons and dicotyledons showed that the tandem duplication events and the segmental duplication events may not only be the primary mechanism of gene family expansion in plant evolution, but may also make a significant contribution to the diversity of gene families [42,43].

### 2.4. Structure and Motif Analysis of EgHsf Genes

To understand the structure of the *EgHsf* gene family, we analyzed the intron–exon structure of 36 *EgHsf* gene members and the domain structure of their protein sequences. As shown in Figure 4, *EgHsf* genes contained conservative internal structures. The phylogenetic analysis between these genes is shown in Figure 4A. Protein sequence analysis identified a total of 14 motifs (Figure 4B). Motif 1, motif 2 and motif 3 represented the characteristic structures of HSF. Motif 1 was mostly combined with motif 2, while in EgHSF5, motif 3 existed alone. In addition, we found that some members had conservative motifs that were absent in other members, such as motif 12, etc., which may imply that these EgHSF protein members had other evolutionary functions to adapt to environmental changes. The prediction of the conserved structural domains of the 36 EgHSF proteins revealed that all EgHSF proteins have highly conserved structural domains (Figure 4C). Interestingly, when we compared the similarity of exon–intron structure between the members of the *EgHsf* gene family, we found that they were similar among *Hsf* gene classes A, B and C (Figure 4D), further verifying the accuracy of previous phylogenetic classification. In addition, we noted that the number of introns and exons in these *EgHsf* genes was inconsistent: 25 of the 36 *EgHsf* genes contained two exons, 8 genes contained three exons, and 1 gene contained only one exon.

Multiple sequence alignment of the DNA-binding domains of 36 EgHSF proteins revealed the conserved amino acid sequences of HSF (Figure 5 and Appendix A), and there was a highly conserved DBD structure (α1-β1-β2-α2-α3-β3-β4) among all members. This fully proved the conservation of the unique structure of this family. The conserved DBD structure among all HSF members also suggested that the majority of HSFs may share some degree of functional similarity.

### 2.5. Promoter Cis-Element Analysis of EgHsf Genes

As non-coding DNA sequences in gene promoters, cis-acting elements are able to regulate the transcription of their associated genes. We submitted the sequence of 1500 base pairs upstream of the *EgHsf* gene transfer initiation site to the PlantCARE database, resulting in the identification of 13 cis-acting elements (Figure 6 and Appendix A). We can clearly see that, in addition to the core cis-acting elements, the other elements include LTR (low-temperature responsiveness), TC-rich repeats (defense and stress responses) and ARE (anaerobic induction), and in terms of hormonal regulation, TCA-element (salicylic acid), CGTCA-motif and TGACG-motif (methyl jasmonate), TGA-element (auxin), ABRE (abscisic acid), etc. In addition, gibberellin (P-box, TATC-box, GARE-motif) was identified, as well as regulatory motifs related to tissue-specific expression (e.g., RY-element and CAT-box), development or cell differentiation (e.g., MSA-like and circadian). Among these, we found that the abscisic acid response element is the most abundant cis-acting element, which is widely present in the *EgHsf* promoter region. Taken together, these results may indicate that the functional expression of *EgHsf* genes is regulated by cis-acting elements associated with plant developmental processes, hormones and abiotic stress responses.

### 2.6. Expression of Hsf Genes in Eucalyptus under Abiotic Stress

To explore the response of *Hsf* genes to abiotic stresses, we chose salt stress and temperature stress to test the response of *Eucalyptus* gene expression (Appendix A). The results show that the expression levels of *EgHsf* genes were regulated in response to salt and temperature. When treated with 100 mM NaCl (Figure 7A), the expression of *EgHsf4*, *EgHsf13*, *EgHsf24* and *EgHsf32* significantly increased at 12 h after treatment, by around 10-fold, compared to the expression level at the start of treatment. Under 200 mM NaCl treatment, the expression of each gene increased to different degrees after 12 h, with the expression of *EgHsf9* already being raised at 6 h. The expression of both *EgHsf4* and *EgHsf24* was more prominent, especially *EgHsf4*, which was maintained at a high expression level after 24 h. These findings suggest that these two *EgHsf* genes (*EgHsf4* and *EgHsf24*) play an important role in plant tolerance to salt stress. At 4 °C treatment (Figure 7B), we found that most *EgHsf* genes were repressed to varying degrees. However, the expression of *EgHsf9*, *EgHsf13*, *EgHsf24* and *EgHsf32* gradually increased after 6 h, with *EgHsf24* reaching a high level at 24 h. At 40 °C treatment, we found that the expression of *EgHsf13*, *EgHsf24* and *EgHsf32* increased at 6 h, with the expression of *EgHsf32* significantly increasing. It might be speculated that *EgHsf24* and *EgHsf32* played an important role in the plant response to temperature stress.

### 2.7. Expression Map and Correlation Analysis of Hsf Genes in Different Tissues

Based on the FPKM (fragments per kilobase of exon model per million mapped fragments) value, the heat map of the *EgHsf* genes was constructed using the program R. To analyze the differences in the expression patterns of *EgHsf*, we included the gene expression data in six tissues, including xylem, immature xylem, phloem, shoot tips, mature leaf and young leaf (Figure 8A). The results suggest that *EgHsf3*, *EgHsf7*, *EgHsf4* and *EgHsf12* had the highest expression in mature leaf, with moderate expression in young leaf. The *EgHsf13* gene had a similarly high expression in mature leaf, but its expression in young leaf was lower compared to *EgHsf3*, *EgHsf7*, *EgHsf4* and *EgHsf12*, suggesting that *EgHsf13* expression was related to leaf development, reaching a maximum at the mature leaf stage. Furthermore, *EgHsf34* was highly expressed in mature xylem compared to other tissues. On the contrary, *EgHsf19* had lower expression in the mature xylem.

We further studied the expression network of various *EgHsf* genes (Appendix A). As shown in Figure 8B, *EgHsf14* was the most active, with a great positive correlation with *EgHsf27*, *EgHsf16*, *EgHsf5*, *EgHsf32*, *EgHsf31*, *EgHsf35* and *EgHsf26*, and a great negative correlation with *EgHsf7*. In addition, the correlation between *EgHsf29* and *EgHsf12* and other surrounding members was not strong (Figure 8C). Interestingly, when we combined the expression heat map, we could even specifically study the *EgHsf* cooperation of a given expression site. For instance, *EgHsf10*, *EgHsf36* and *EgHsf28* had high expression in the phloem. There was a great positive correlation between *EgHsf10* and *EgHsf36*, but the correlation between them and *EgHsf28* was not obvious. This may provide some basic reference for further revealing the interaction between *EgHsf* genes.

## 3. Discussion

*Hsf* plays a role in protecting plants from deviations to normal growth [14,15], as well as in coping with heat and cold stress and maintaining cellular homeostasis [20]. Based on the importance of *Hsf* genes for the study of abiotic stress in plants [44], they have been found in many different plants, such as wheat, grape, pepper, mullein, cucumber [29,30,31,32,33], etc. As one of the more dominant economic woody species, *Eucalyptus* is a good model for studying abiotic stresses; however, studies on the *Eucalyptus Hsf* gene family have not been published.

A total of 36 *Hsf* genes were identified in this study to reveal the role of *Hsf* in terms of the response to abiotic stress in *Eucalyptus*. Moreover, it is obvious that the *Hsf* genes are unevenly distributed on the 11 chromosomes. Among them, chromosome 3 has the highest number of *Hsf* genes, and we can reasonably guess that chromosome 3 may also be one of the key chromosomes for plants to cope with abiotic stresses. To investigate the evolutionary features of these genes and the evolutionary relationships with HSF proteins, we constructed phylogenetic trees using HSF proteins from *Eucalyptus*, *Arabidopsis* and rice, and thus classified *EgHsf* into three categories. After analyzing the structural domains and conserved sequences of these 36 members, we found some similarities in exon structures among all *EgHsf* gene family members, in which we found that some of the structural domains are conserved, such as motif 12, which may indicate that these *EgHsf* members are evolving to adapt to their environment. As an important component of gene expression, cis-acting elements play an important role in plant growth and development with respect to environmental adaptation. In the present study, we identified many regulatory motifs in the putative promoter region of the *EgHsf* gene, and the most notable was the widely distributed abscisic acid (ABA) response element, which, as an important plant hormone, regulates plant growth and development and the stress response and plays an important role in a variety of physiological processes in plants [45]. Interestingly, it has been shown that maize-related *ZmHsf* gene expression is downregulated after ABA treatment [46]. Analysis of cis-acting elements suggests that *EgHsf* genes may be involved in various biological processes through the regulation of multiple target genes related to growth and development, metabolism and abiotic stresses. Previous studies by Nover et al. (2001) demonstrated that, in most *Hsf*genes, there was a conserved intron insertion, so the DBD structure was separated in two exons [47]. However, the *EgHsf**5* that belonged to the *EgHsf**A1* subclass did not have an intron, which might have been a lateral manifestation of the unique dominance of members of the *Hsf**A* group in terms of heat stress [48]. Additionally, multiple sequence comparisons of the DNA-binding domains of these EgHSF proteins revealed a highly conserved DBD structure (α1-β1-β2-α2-α3-β3-β4) among all members, which is a strong indication of the degree of conservation of the unique structure of this family. Additionally, from the synteny analysis, the effect of the whole-genome duplication (WGD) event on the expansion of *EgHsf* among various plants was profoundly long-term and gigantic [49,50]. In order to further infer the evolution of the *EgHsf* gene family, we calculated the Ka/Ks value of *Hsf* genes (Appendix A). The most homologous Ka/Ks value was smaller than 1, which suggested that the Eg*Hsf* gene family had experienced great changes in its long-term evolutionary process under great purification selection pressures.

According to the cis-acting element analysis (Appendix A), cis-acting elements associated with salt and temperature stress, such as ABRE (abscisic acid responsiveness) and LTR (low-temperature responsiveness), were distributed in the promoter regions of many *EgHsf* genes, such as *EgHsf24* and *EgHsf32*. From the analysis of *EgHsf* gene expression patterns, we found that many *EgHsf* homologous gene pairs were expressed at similar levels and may exhibit functional redundancy. Additionally, *EgHsf3*, *EgHsf7*, *EgHsf4* and *EgHsf12* had the highest expression in mature leaves, which may indicate that these genes have an important role to play in maintaining the balance between mature leaves and the external environment. Additionally, in xylem and siliques, we can see that there is a difference in expression patterns. From the differences in the expression patterns of *EgHsf* genes, the differential expression of *EgHsf* implied that they had stage-specific expression patterns during xylem and phloem development. The expression pattern of the *EgHsf* among three tissues is likely to be associated with their functions in vascular organ development. Further study by qRT-PCR under salt stress revealed that the expression of most of the selected *EgHsf* genes increased with time after treatment, and among them, *EgHsf4* was able to maintain a high expression level for a longer period of time. On the other hand, under low-temperature treatment, we found that the majority of *EgHsf* genes were repressed to different degrees, but with time, the expression of most of the selected *EgHsf* genes gradually increased after 6 h, with *EgHsf24* reaching a higher level at 24 h. Additionally, under high-temperature stress, we found that the expression of both *EgHsf32* and *EgHsf24* increased to different degrees over time, and in this experiment, we also noticed that some genes of the *EgHsfB* subclass are well-adapted to abiotic stresses (*EgHsf24* and *EgHsf32*). Due to the structural similarity of these two classes, we speculated that this may be related to the independent regulation of plant metabolism by *HsfA* [51]. On the basis of these findings, it is possible to predict the potential role of the *EgHsf* gene in response to salt and temperature conditions (Figure 9). It was further shown by the expression network of *EgHsf* genes that there may be some interactions between different genes, which could provide the basis for the subsequent study of the effects of gene interactions on abiotic stresses.

## 4. Materials and Methods

### 4.1. Plant Material and Data Sources

Seedlings of Eucalyptus clone Eg5 were provided by Fujian Academy of Forestry. Under outdoor conditions, they were cultivated in red soil with a soil organic matter content of 2.57~6.07% at pH 5 for 10 months. The average annual temperature in the growth area was 16~20 °C, with an average annual precipitation of 900~2100 mm and an annual relative humidity of approximately 77%. The whole-genome data of Eucalyptus were downloaded from the National Center for Biotechnology Information (NCBI, https://www.ncbi.nlm.nih.gov/, accessed on 1 January 2022).

### 4.2. Identification and Sequence Analysis of EgHSF Protein

The DBD domain (PF00447) of HSF was obtained from Pfam (http://pfam.xfam.org/, accessed on 2 January 2022), and all protein sequences of *Eucalyptus* were screened in HMMER-3.3. The standard e-value was <10^−5^. To verify that these candidates were *Hsfs*, we used the NCBI Conserved Domain Database (https://www.ncbi.nlm.nih.gov/cdd/, accessed on 2 January 2022) (E-value < 1 × 10^−5^, other parameters set to defaults) to filter the HSF domain sequence. The molecular weight and isoelectric points of deduced proteins were analyzed by ExPASy (https://web.expasy.org/protparam, accessed on 2 January 2022). Subcellular localization was predicted using Cell-PLoc 2.0 (http://www.csbio.sjtu.edu.cn/bioinf/Cell-PLoc-2/, accessed on 2 January 2022) [52].

### 4.3. Classification of EgHSF Protein Members and Construction of Phylogenetic Tree

The phylogenetic tree of related proteins was constructed by using the MEGA 11 software using the maximum likelihood method, and the verification parameter bootstrap was set to 1000. The results of the evolutionary tree were visualized by Evolview (http://www.evolgenius.info/evolview/#/, accessed on 3 January 2022) for post-processing.

### 4.4. Chromosomal Distribution and Synteny Analysis of the EgHsf Genes

The identified *Hsf* genes were mapped to *Eucalyptus* chromosomes by comparing them to *Eucalyptus* genome data, and analysis of their collinearity was carried out with TBtools software [53]. In order to reveal the common relationship between homologous *Hsf* genes obtained in *Eucalyptus* and other selected species, we downloaded the whole-genome sequences and gene annotation files of seven species (*Arabidopsis thaliana*, *Zea mays*, *Oryza sativa*, *Vitis vinifera*, *Populus trichocarpa*, *Hordeum vulgare* and *Solanum lycopersicum*) from the plant genome database. The commonness analysis atlas was constructed using TBtools software. The Ka/Ks_Calculator in TBTools software was used to calculate the nonsynonymous substitution rate (Ka), synonymous substitution rate (Ks) and Ka/Ks to investigate selection pressures.

### 4.5. Structure and Motif Analysis of EgHSFs

The conserved motifs of *Eucalyptus* HSF proteins were analyzed by MEME (http://meme-suite.org/, accessed on 4 January 2022) [54]. Multiple sequence alignment of *Eucalyptus* HSF protein sequences was carried out by Jalview software (V2.11.1.5) (http://www.jalview.org/, accessed on 4 January 2022) [55] to analyze the conserved domain of HSF. The Jalview output was transferred to JPred (http://www.compbio.dundee.ac.uk/jabaws, accessed on 4 January 2022), using default parameters for the prediction of protein secondary structure [56].

### 4.6. Promoter Cis-Element Analysis of EgHsf Genes

PlantCare [57] (http://bioinformatics.psb.ugent.be/webtools/plantcare/html/, accessed on 5 January 2022) was used to predict the cis-acting elements in the putative promoter region of *EgHsf* genes. The results were visualized using TBtools [53] and OmicStudio tools (https://www.omicstudio.cn, accessed on 5 January 2022).

### 4.7. Heat Map Network Analysis of Hsf Gene Expression and Correlation Cluster Markers

The expression data of *Hsf* genes of *E. grandis* in various tissues and under diverse environmental conditions were derived from the Phytozome database. The expression data of these *Hsf* genes were analyzed by R, and a gene expression heat map was constructed. The low-salt (100 mM NaCl) and high-salt (200 mM NaCl) treatments were carried out for the same periods of time (0, 6, 12 and 24 h), and the control seedlings were treated with distilled water. Similarly, the low-temperature (4 °C) treatment, high-temperature (40 °C) treatment and control treatment at room temperature (25 °C) were carried out. The young leaves of *E. grandis* were collected at 0, 6, 12 and 24 h of treatment. The samples were stored at −80 °C for subsequent analysis. Five individual seedlings were used in each stress treatment, and the experiment was repeated at least three times. The collected leaves were ground to extract RNA, and a real-time fluorescence quantitative PCR (qRT-PCR) experiment was carried out for monitoring the gene expression level. A correlation clustering labeled heat map was drawn by the Spearman correlation algorithm [58].

### 4.8. RNA Extraction and qRT–PCR Analysis

Total RNA was extracted from the control and the stress-treated samples with an RNA Extraction Kit (Omega Bio-TEK, Shanghai, China). EasyScript*^®^* One-step gDNA Removal and cDNA Synthesis SuperMix were used to synthesize cDNA according to the manufacturer’s instructions (Transgen, Beijing, China). Quantitative RT-PCR was performed with TransStart^®^ top green qPCR SuperMix (Transgen, Beijing, China). Actin was used as an internal reference gene and the primers used are listed in Appendix A. Relative transcript abundance was calculated by the comparative 2^−ΔΔCT^ method [59]. All quantitative PCRs were performed with three biological repeats and three technical replicates.

## 5. Conclusions

In this study, a total of 36 *EgHsf* genes were identified in *Eucalyptus*. *EgHsf* members were classified into three classes by evolutionary analysis, and the same subclasses tended to have similar exon–intron structure, protein structure and motifs. The differential expression of *EgHsf* gene members in different tissues of *Eucalyptus* was found to be likely due to their functional diversity. The expression of both *EgHsf* classes was more outstanding under salt stress and temperature stress. This suggests that individual *EgHsf* gene members may play unique roles under different abiotic stress conditions. In the context of the increasingly severe greenhouse effect and complex, gradual environmental changes, this study provides more comprehensive information on the function of the *EgHsf* gene in *Eucalyptus* and lays a theoretical foundation for further research on the mechanism of action of the relevant genes.

## Figures and Tables

**Figure 1 ijms-23-08044-f001:**
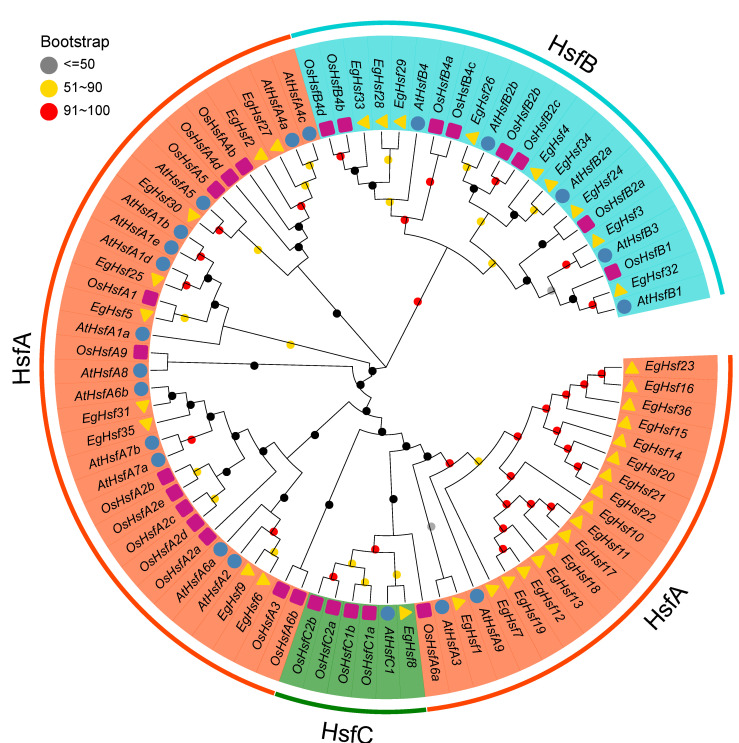
Phylogenetic analysis of full-length HSF protein sequences from *E. grandis* (Eg), *A. thaliana* (At) and *Oryza sativa* (Os). The number on the branch denotes the reliability of the node based on 1000 iterations of Bootstrap verification; branches of different classes have altered colors, each standing for a different subfamily.

**Figure 2 ijms-23-08044-f002:**
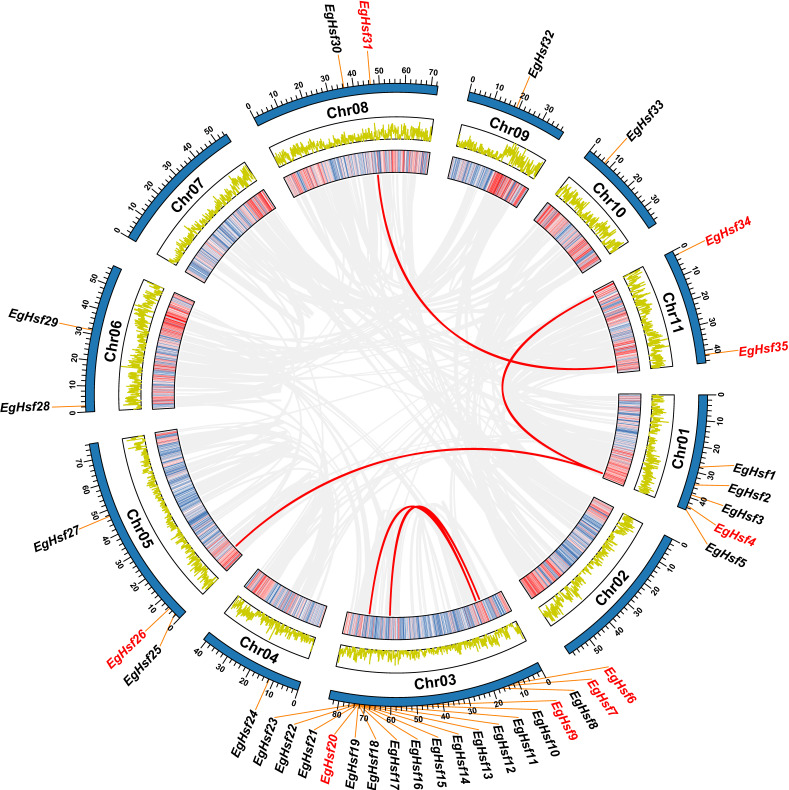
Chromosomal distribution and inter-chromosomal relationship of *EgHsf* genes. The two rings in the middle represent the gene density per chromosome, and the grey line represents the collinear block in the genome, while the red line represents the repeated *EgHsf* gene pair. *EgHsf* genes marked in red have collinearity, while *EgHsf* genes marked in black lack collinearity. TBtools was used for data processing.

**Figure 3 ijms-23-08044-f003:**
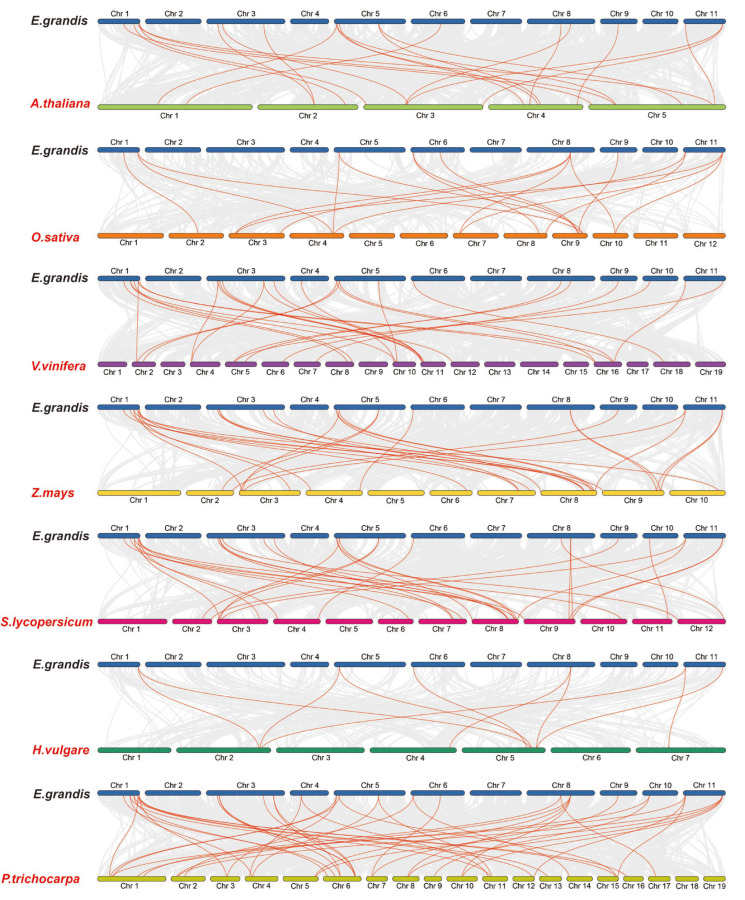
Homology analysis between eucalyptus genome and seven plant genomes (*Arabidopsis thaliana*, *Zea mays*, *Oryza sativa*, *Vitis vinifera*, *Populus Tomentosa*, *Hordeum vulgare* and *Solanum lycopersicum*). The grey lines symbolize the aligned blocks between paired genomes, and the red lines stand for collinear *EgHsf* gene pairs.

**Figure 4 ijms-23-08044-f004:**
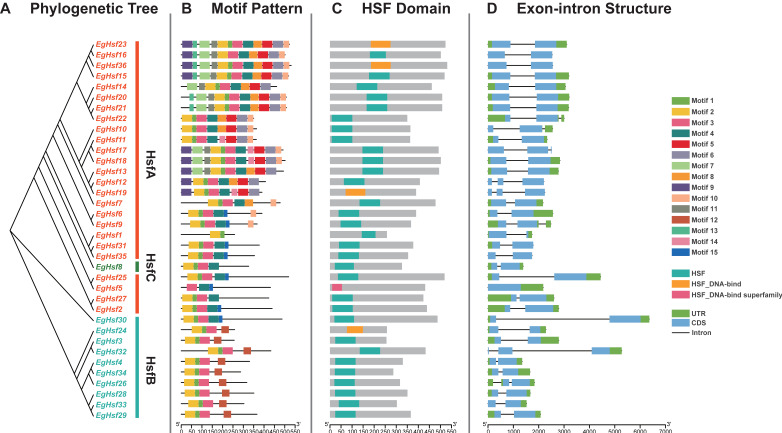
Structure of *EgHsf* gene members. (**A**) Phylogenetic relationships among *EgHsf* members. (**B**) Distribution of conserved motifs of the EgHSF protein. A total of 15 motifs were identified. The scale at the bottom shows the length of the protein, and the sequence identity of each conserved motif is marked on the right. (**C**) Predicted conserved structural domains of EgHSF proteins. Grey bars represent the length of each protein sequence, and conserved domains are indicated by colored boxes. (**D**) Exon–intron structure of the *EgHsf* genes. Yellow boxes indicate exons (CDS), black lines indicate introns and green boxes indicate 5′ and 3′ untranslated regions.

**Figure 5 ijms-23-08044-f005:**
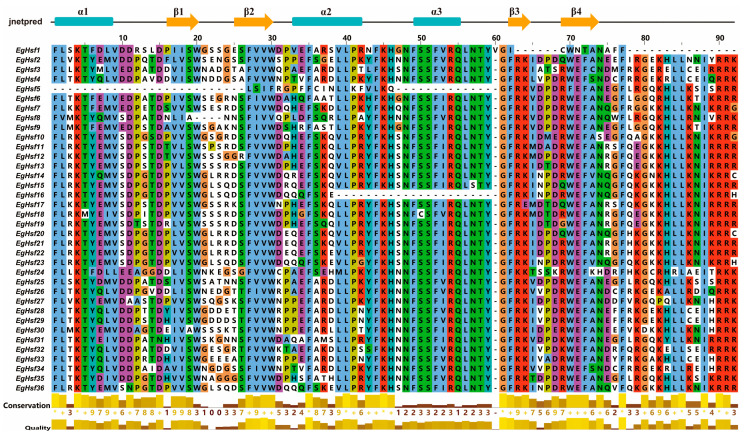
Conversed amino acid sequence in DNA-binding domain of EgHSFs. “*” means that the amino acid residues at this position are absolutely conserved after multiple sequence alignment.

**Figure 6 ijms-23-08044-f006:**
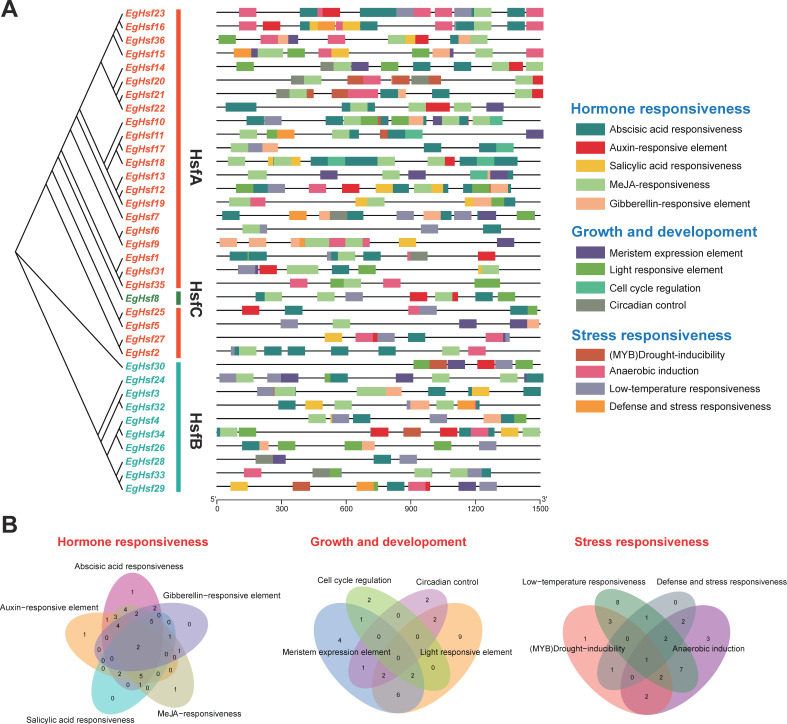
The cis-acting element on the putative promoter of the *E**gHsf* genes. (**A**) Distribution of cis-acting elements identified in the 1500 bp upstream promoter region of *EgHsf* genes. (**B**) Venn diagram of various cis-acting elements.

**Figure 7 ijms-23-08044-f007:**
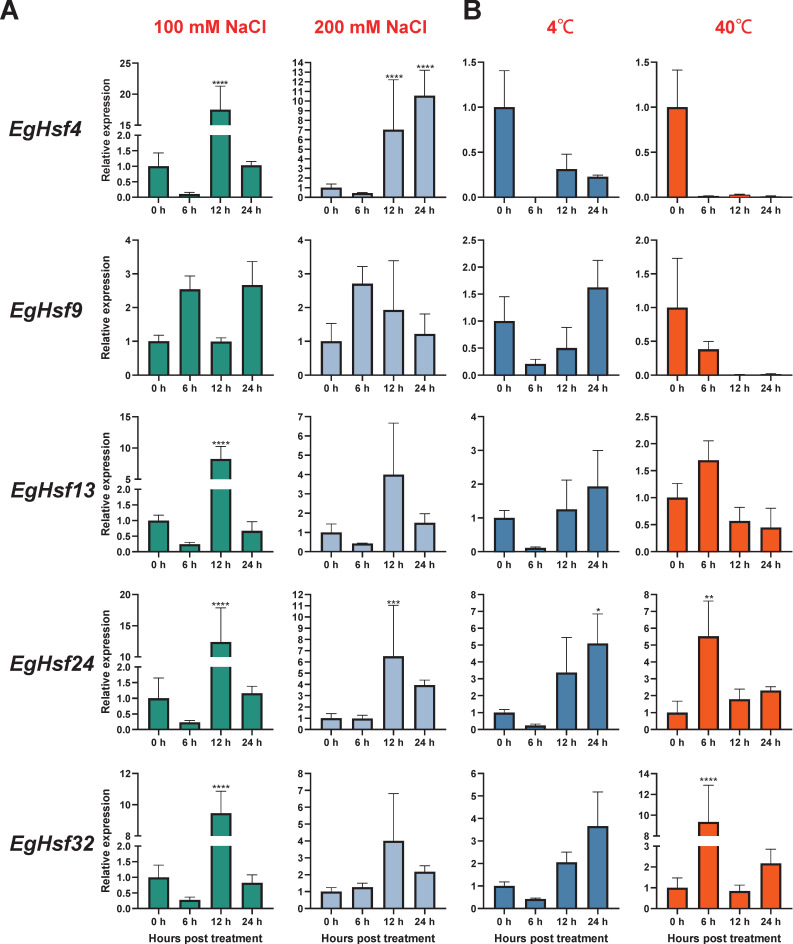
Expression profile of *EgHsf* genes in *eucalyptus* responding to salt and temperature stresses tested by qRT-PCR. (**A**) The relative gene expression levels under low salt (100 mL NaCl) and high salt (200 mL NaCl) treatments for the same time periods (0, 6, 12, 24 h). Control seedlings were treated with distilled water. (**B**) Relative gene expression levels under low temperature (4 °C), high temperature (40 °C) and control (25 °C). (* *p* < 0.05, ** *p* < 0.01, *** *p* < 0.0005, **** *p* < 0.0001).

**Figure 8 ijms-23-08044-f008:**
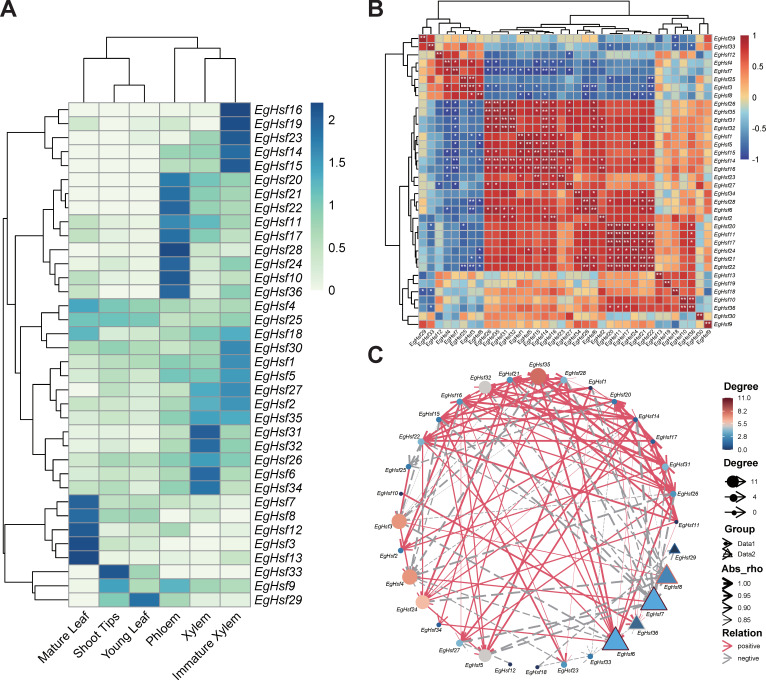
Analysis of the expression patterns and networking of 36 *EgHsf* genes in Eucalyptus. (**A**) Gene expression patterns in six tissues, including xylem, immature xylem, phloem, shoot tips, mature leaf and young leaf. The color in the upper right corner represents the expression level: red signifies a high expression level, while blue signifies a low expression level. (**B**) Correlation of gene expression. The darker the color in the figure, the stronger the correlation. Red denotes a positive correlation, and blue represents a negative correlation. The asterisk records the significance of the correlation value. The more asterisks, the greater the significance. The picture is diagonally symmetric for a single table, and the diagonal number is always 1 (the correlation between the same genes was 100%, * represents < 0.05, and ** represents < 0.01). (**C**) Gene network. Red line represents a positive connection with surrounding genes: the denser and thicker the line, the higher the significance. The grey line represents the negative correlation effect, and the color from blue to red shows the strength of the correlation.

**Figure 9 ijms-23-08044-f009:**
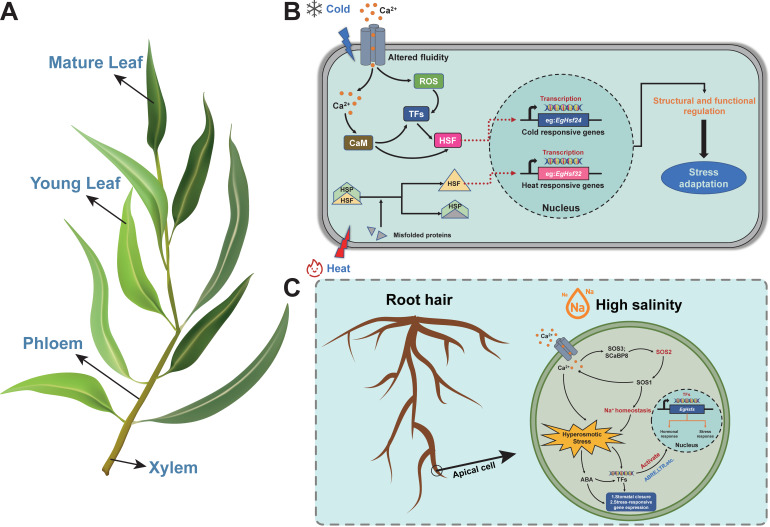
Abiotic stress (salt and temperature) response mechanisms of the *Hsf* gene in *Eucalyptus*. (**A**) Schematic diagram of the morphology of the *Eucalyptus* branch. The diagram includes mature leaves, young leaves, phloem and xylem. These parts (including the roots) are the key parts of *Eucalyptus* in relation to abiotic stress perception and response. (**B**) Schematic diagram of the temperature stress response mechanism and signaling for this experiment. Low-temperature stress may lead to elevated levels of calcium ions and reactive oxygen species in the cytoplasm of the cell, which are recognized by TFs or act directly on the responding genes before entering the nucleus to participate in the gene expression response to cold stress. In the present experiment, *EgHsf24* was the best performer in response to the corresponding stress, while high temperature caused the denaturation of proteins in the cell, resulting in misfolding, which led to the activation of sensory genes, thus entering the nucleus to cause expression in response to heat stress. *EgHsf32* performed well in response to heat stress in this experiment. The final result was the enhancement of plant tolerance to temperature stress. (**C**) Schematic diagram of the mechanism of salt stress response and signaling in this experiment. Under high-salt conditions, the receptor module on the cell membrane induces calcium ions to enter the cell, and the calcium signal activates the SOS3-SOS2-SOS1 pathway, at which time SOS2 is activated (red), thus gradually stabilizing the sodium ion concentration. The osmotically induced stress effect generated by the increased intracellular calcium ion concentration causes the accumulation of ABA in the plant body, forming two response pathways: (1) undergoing a series of mechanistic changes to close the plant stomata, which is one of the ways to cope with drought stress, and (2) undergoing mechanistic changes to bind with TFs, thus causing the expression of stress response genes and activating the corresponding cis-acting elements (such as ABRE, LTR, etc.), ultimately enhancing the tolerance of plants to salt stress. *EgH*sf4 performed well in response to salt stress in this experiment. CaM, Ca^2+^-calmodulin; ROS, reactive oxygen species; TFs, transcription factors; SOS, salt over-sensitive; ABA, abscisic acid; ABRE, abscisic acid responsiveness; LTR, low-temperature responsiveness. (Adapted from Zhang, H. et al. (2021) [44] and Guo, M. et al. (2016) [22]).

**Table 1 ijms-23-08044-t001:** Detailed information on 36 *EgHsf* genes of *Eucalyptus* and their encoded proteins.

Number	Gene	Member	Chromosome Location	Gene Length(nt)	Protein Molecular Weight (MW, Da)	Protein Isoelectric Point (pI)	Predicted Subcellular Location
1	Eucgr.A01544	*EgHsf1*	Chr01	1737	27,565	4.75	Nucleus
2	Eucgr.A01948	*EgHsf2*	Chr01	2791	49,802	5.19	Nucleus
3	Eucgr.A02338	*EgHsf3*	Chr01	2793	29,215	8.25	Nucleus
4	Eucgr.A02804	*EgHsf4*	Chr01	1349	35,566	4.78	Nucleus
5	Eucgr.A02976	*EgHsf5*	Chr01	2179	47,615	5.67	Nucleus
6	Eucgr.C00664	*EgHsf6*	Chr03	2559	43,514	4.94	Nucleus
7	Eucgr.C00873	*EgHsf7*	Chr03	2170	52,813	4.61	Nucleus
8	Eucgr.C01043	*EgHsf8*	Chr03	1389	35,714	5.31	Nucleus
9	Eucgr.C03056	*EgHsf9*	Chr03	2482	41,266	4.56	Nucleus
10	Eucgr.C03424	*EgHsf10*	Chr03	2550	41,407	5.78	Nucleus
11	Eucgr.C03431	*EgHsf11*	Chr03	2333	41,444	5.99	Nucleus
12	Eucgr.C03433	*EgHsf12*	Chr03	2205	45,858	6.11	Nucleus
13	Eucgr.C03434	*EgHsf13*	Chr03	2780	54,933	5.21	Nucleus
14	Eucgr.C03435	*EgHsf14*	Chr03	3056	51,533	5.92	Nucleus
15	Eucgr.C03440	*EgHsf15*	Chr03	3191	57,928	6.06	Nucleus
16	Eucgr.C03441	*EgHsf16*	Chr03	2534	55,594	5.76	Nucleus
17	Eucgr.C03447	*EgHsf17*	Chr03	2508	54,914	5.24	Nucleus
18	Eucgr.C03449	*EgHsf18*	Chr03	2838	55,702	5.05	Nucleus
19	Eucgr.C03452	*EgHsf19*	Chr03	2243	44,186	5.84	Nucleus
20	Eucgr.C03454	*EgHsf20*	Chr03	3203	57,199	5.91	Nucleus
21	Eucgr.C03456	*EgHsf21*	Chr03	3183	57,348	6.07	Nucleus
22	Eucgr.C03457	*EgHsf22*	Chr03	3004	40,621	5.72	Nucleus
23	Eucgr.C03465	*EgHsf23*	Chr03	3107	58,095	5.98	Nucleus
24	Eucgr.D00627	*EgHsf24*	Chr04	2286	29,766	6.33	Nucleus
25	Eucgr.E00253	*EgHsf25*	Chr05	4442	56,475	4.72	Nucleus
26	Eucgr.E00555	*EgHsf26*	Chr05	1834	33,858	9.41	Nucleus
27	Eucgr.E02813	*EgHsf27*	Chr05	2604	47,555	5.09	Nucleus
28	Eucgr.F00215	*EgHsf28*	Chr06	1668	39,371	7.11	Nucleus
29	Eucgr.F02327	*EgHsf29*	Chr06	2075	40,710	8.44	Nucleus
30	Eucgr.H02675	*EgHsf30*	Chr08	6363	54,320	5.93	Nucleus
31	Eucgr.H03412	*EgHsf31*	Chr08	1784	42,895	5.35	Nucleus
32	Eucgr.I00929	*EgHsf32*	Chr09	5275	46,303	6.68	Nucleus
33	Eucgr.J00680	*EgHsf33*	Chr10	1523	34,909	7.11	Nucleus
34	Eucgr.K00238	*EgHsf34*	Chr11	1658	31,565	9.33	Nucleus
35	Eucgr.K03325	*EgHsf35*	Chr11	1740	39,837	5.33	Nucleus
36	Eucgr.L01331	*EgHsf36*	scaffold_170	2553	59,378	6.31	Nucleus

## Data Availability

The datasets presented in this study can be found in online repositories. The repository/repositories and accession number(s) can be found below: https://www.ncbi.nlm.nih.gov/ (accessed on 1 January 2022), PRJNA49047.

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
