# Peer review of "Genome-Wide Identification of Eucalyptus Heat Shock Transcription Factor Family and Their Transcriptional Analysis under Salt and Temperature Stresses"

_ijms, 2022, doi:10.3390/ijms23148044_

Round 1
Reviewer 1 Report
The manuscript “Genome-wide identification of Eucalyptus heat shock transcription factor family and their transcriptional analysis under salt and temperature stresses” aimed provide basic information on the members of the Hsf gene family in Eucalyptus and lays the foundation for the functional identification of related genes and further investigation on their biological functions in plant stress regulation.
The Authors present an interesting manuscript and MS brings new elements to existing knowledge about Hsf gene family in Eucalyptus.
The manuscript is prepared professionally. It includes a well-crafted abstract and an exhaustive introduction that justifies the research undertaken. The introduction points to the deficiencies in the literature on the subject. The aim is clearly defined. Modern analytical methods were used in the research. The discussion of the results is well prepared. The conclusions are well-defined. The illustrative material is appropriate.
In my opinion, the manuscript after corrections, will be suitable for publication in a journal.
Detailed comments:
Introduction - The introduction is enough in my opinion. However the authors may have thing add an introduction sentence about importance of forest trees in introduction part and after that they can talk about Eucalyptus.
I prepared below one.
Forest trees benefit people by removing carbon from the atmosphere, cooling our neighborhoods, and filtering our water and air. Forests are also provide shelter, livelihoods, water, food and fuel security (Han et al., 2021;
Han, S.H.; Kim, S.; Chang, H.N.; Kim, H.J.; An, J.; Son, Y. Fine root biomass and production regarding root diameter in Pinus densiflora and Quercus serrata forests: Soil depth effects and the relationship with net primary production. Turk. J. Agric. For. 2021, 45 (1), 46-54. DOI 10.3906/tar-1912-13.
Structural and ecological characteristics of mixed broadleaved old-growth forest (Biogradska Gora-Montenegro). Turk. J. Agric. For. 44 (4), 428-438. DOI 10.3906/tar-2003-103.
The authors may have include more information about Genome wide identification because in introduction there is no much information about it.
Author Response
Reviewer 1:
The manuscript “Genome-wide identification of Eucalyptus heat shock transcription factor family and their transcriptional analysis under salt and temperature stresses” aimed to provide basic information on the members of the Hsf gene family in Eucalyptus and lays the foundation for the functional identification of related genes and further investigation on their biological functions in plant stress regulation.
The Authors present an interesting manuscript and MS brings new elements to existing knowledge about the Hsf gene family in Eucalyptus.
The manuscript is prepared professionally. It includes a well-crafted abstract and an exhaustive introduction that justifies the research undertaken. The introduction points to the deficiencies in the literature on the subject. The aim is clearly defined. Modern analytical methods were used in the research. The discussion of the results is well prepared. The conclusions are well-defined. The illustrative material is appropriate.
In my opinion, the manuscript after corrections will be suitable for publication in a journal.
Detailed comments:
Introduction - The introduction is enough in my opinion. However, the authors may have thing add an introduction sentence about the importance of forest trees in the introduction part and after that, they can talk about Eucalyptus.
Thank you very much for your suggestion. We have carefully considered and placed the introductory sentence about the importance of describing the forest trees[1, 2] at the beginning of the introduction so that the reader can fully understand the relevant background. (Line 42-44).
- Han, S.H.K., S.; Chang, H.N.; Kim, H.J.; An, J.; Son, Y., Fine root biomass and production regarding root diameter in Pinusdensiflora and Quercusserrata forests: Soil depth effects and the relationship with net primary production. Turkish Journal of Agriculture and Forestry, 2021. 45(1).
- Curovic, M., et al., Structural and ecological characteristics of mixed broadleaved old-growth forest(Biogradska Gora - Montenegro). Turkish Journal of Agriculture and Forestry, 2020. 44(4): p. 428-438.

Reviewer 2 Report
In this manuscript, the authors did genome-wide identification of the Eucalyptus heat shock transcription factor family and their transcriptional analysis under salt and temperature stresses. In this study, the authors used bioinformatics tools to analyze and identify Eucalyptus Hsf genes, their chromosomal localization, and their structure. The phylogenetic relationship and conserved domains of their encoded proteins were further analyzed. A total of 36 Hsf genes were identified and authenticated from Eucalyptus, which were scattered on 11 chromosomes. They could be classified into three classes (A, B, and C). The expression profiles of five representative Hsf genes EgHsf4, EgHsf9, EgHsf13, EgHsf24, and EgHsf32, under salt and temperature stresses, were examined by qRT-PCR. The results showed that the expression pattern of class B genes (EgHsf4, EgHsf24, and EgHsf32) was relative more tolerant to abiotic stresses than that of class A genes (EgHsf9 and EgHsf13). However, the expressions of all tested Hsf genes in six tissues showed at different levels. The authors investigated the connection between genes, and the results suggested that there might be synergistic effects between different Hsf genes in response to abiotic stresses. The authors concluded that the Hsf gene family played an essential role in Eucalyptus's growth and developmental processes and could be vital for maintaining cell homeostasis against external stresses.
The manuscript Is written well scientifically; however, I have a few suggestions for the authors to improve the manuscript:
1. It would be better if the authors tried to functionally validate at least one gene they found in this study, any abiotic stress, or they could do promoter analysis using any reporter gene.
2. It would be better if the authors made one hypothetical figure for the discussion section and depicted the finding of this study.
3. Manuscript line numbers were missing.
4. In Section 2.4, the structure of the gene is a protein, so it should not be italic. Make all the genes italic, not protein.
5. The introduction is short. The author should include recent genome-wide studies such as:
a. Genome-wide identification and expression pattern analysis of the KCS gene family in barley.
b. Genome-Wide Analysis and Characterization of the Proline-Rich Extensin-like Receptor Kinases (PERKs) Gene Family Reveals Their Role in Different Developmental Stages and Stress Conditions in Wheat (Triticum aestivum L.)
c. Genome-wide identification and characterization of abiotic stress-responsive lncRNAs in Capsicum annuum.
d. Genome-wide identification and functional characterization of natural antisense transcripts in Salvia miltiorrhiza.
Author Response
Reviewer 2:
In this manuscript, the authors did genome-wide identification of the Eucalyptus heat shock transcription factor family and their transcriptional analysis under salt and temperature stresses. In this study, the authors used bioinformatics tools to analyze and identify Eucalyptus Hsf genes, their chromosomal localization, and their structure. The phylogenetic relationship and conserved domains of their encoded proteins were further analyzed. A total of 36 Hsf genes were identified and authenticated from Eucalyptus, which were scattered on 11 chromosomes. They could be classified into three classes (A, B, and C). The expression profiles of five representative Hsf genes EgHsf4, EgHsf9, EgHsf13, EgHsf24, and EgHsf32, under salt and temperature stresses, were examined by qRT-PCR. The results showed that the expression pattern of class B genes (EgHsf4, EgHsf24, and EgHsf32) was relative more tolerant to abiotic stresses than that of class A genes (EgHsf9 and EgHsf13). However, the expressions of all tested Hsf genes in six tissues showed at different levels. The authors investigated the connection between genes, and the results suggested that there might be synergistic effects between different Hsf genes in response to abiotic stresses. The authors concluded that the Hsf gene family played an essential role in Eucalyptus's growth and developmental processes and could be vital for maintaining cell homeostasis against external stresses.
The manuscript Is written well scientifically; however, I have a few suggestions for the authors to improve the manuscript:
1.
It would be better if the authors tried to functionally validate at least one gene they found in this study, any abiotic stress, or they could do promoter analysis using any reporter gene.
We appreciate and respect your suggestion and, after consideration, we have included a promoter analysis in the manuscript (Section 2.5) and conclude that the functional expression of the Eucalyptus Hsf gene is regulated by cis-acting elements related to plant developmental processes, hormones, and abiotic stress responses. (Line 216-232).
We have also added a description of this conclusion in the "Summary" and "Conclusions" sections. And two references have been added to the "Conclusion" section[1, 2] (Line 24-27 and Line 316-326).
2. It would be better if the authors made one hypothetical figure for the discussion section and depicted the finding of this study.
Thank you very much for your comment, and after discussions with other authors and an extensive literature review, we have created a related figure combining the results obtained in this paper.
3. Manuscript line numbers were missing.
Thanks for the reminder, all line numbers have been added.
4. In Section 2.4, the structure of the gene is a protein, so it should not be italic. Make all the genes italic, not protein.
Thank you for pointing out the problem with this paragraph, and we apologize for the distress caused by the improper forward italics, which we have checked and made clearer, clearly indicating the naming of proteins and genes. For example: “EgHsf gene members” (line 179 and 196) and “EgHSF proteins members”(line 186).
5. The introduction is short. The author should include recent genome-wide studies such as:
- Genome-wide identification and expression pattern analysis of the KCS gene family in barley.
- Genome-Wide Analysis and Characterization of the Proline-Rich Extensin-like Receptor Kinases (PERKs) Gene Family Reveals Their Role in Different Developmental Stages and Stress Conditions in Wheat (Triticum aestivumL.)
- Genome-wide identification and characterization of abiotic stress-responsive lncRNAs in Capsicum annuum.
- Genome-wide identification and functional characterization of natural antisense transcripts in Salvia miltiorrhiza.
Thank you for your constructive comments on the preface, we have followed your comments and made appropriate changes to the preface, and introduced the purpose of the genome-wide study, leading to the recent genome-wide study[3-7]. (Line 84-88)
- Wang, J., et al., A Novel Heat Shock Transcription Factor (ZmHsf08) Negatively Regulates Salt and Drought Stress Responses in Maize. Int J Mol Sci, 2021. 22(21).
- Chen, K., et al., Abscisic acid dynamics, signaling, and functions in plants. J Integr Plant Biol, 2020. 62(1): p. 25-54.
- Baruah, P.M., et al., Genome wide identification and characterization of abiotic stress responsive lncRNAs in Capsicum annuum. Plant Physiol Biochem, 2021. 162: p. 221-236.
- Jiang, M., et al., Genome-wide identification and functional characterization of natural antisense transcripts in Salvia miltiorrhiza. Sci Rep, 2021. 11(1): p. 4769.
- Kesawat, M.S., et al., Genome-Wide Analysis and Characterization of the Proline-Rich Extensin-like Receptor Kinases (PERKs) Gene Family Reveals Their Role in Different Developmental Stages and Stress Conditions in Wheat (Triticum aestivum L.). Plants (Basel), 2022. 11(4).
- Luo, J., et al., Genome-Wide Identification of the LHC Gene Family in Kiwifruit and Regulatory Role of AcLhcb3.1/3.2 for Chlorophyll a Content. Int J Mol Sci, 2022. 23(12).
- Tong, T., et al., Genome-wide identification and expression pattern analysis of the KCS gene family in barley. Plant Growth Regulation, 2020. 93(1): p. 89-103.

Round 2
Reviewer 1 Report
Dear Editor,
I carefully checked the revised version. The authors made all substantial changes and additions on revised version. I believe that the paper is now ready for publication in IJMS.
Reviewer 2 Report
I am happy with the author's comments. The manuscript looks improved now and can be accepted in its current format.